# Associations among Active Commuting to School and Prevalence of Obesity in Adolescents: A Systematic Review

**DOI:** 10.3390/ijerph191710852

**Published:** 2022-08-31

**Authors:** Evelyn Martin-Moraleda, Sandy Mandic, Ana Queralt, Cristina Romero-Blanco, Susana Aznar

**Affiliations:** 1PAFS Research Group, Faculty of Sports Sciences, University of Castilla-La Mancha, 45004 Toledo, Spain; 2Faculty of Health and Environmental Sciences, School of Sport and Recreation, Auckland University of Technology, Auckland 1010, New Zealand; 3AGILE Research Ltd., Wellington 6012, New Zealand; 4Department of Nursing, University of Valencia, 46010 Valencia, Spain; 5PAFS Research Group, Faculty of Nursing, University of Castilla-La Mancha, 13071 Ciudad Real, Spain; 6CIBER of Frailty and Healthy Aging (CIBERFES), 28029 Madrid, Spain

**Keywords:** adolescents, active commuting, obesity, overweight, school

## Abstract

Active commuting to school (ACS) seems to be one of the means to increase physical activity (PA) levels in youth, but it is unclear if ACS reduces the prevalence of obesity, protecting and improving their health. Most of the previous research has been conducted on children or youth (i.e., children with adolescents together), and there is a paucity of research in adolescents only. The purpose of this review was to assess the association between ACS with overweight/obesity parameters in adolescents aged 11 to 19 years. We used PubMed, WOS and SPORTDiscus as electronics databases. All steps of the process followed the recommendations of the PRISMA flow-diagram. Fifteen articles (68.18%) found a consistent association between ACS and body composition and seven studies (31.82%) showed no differences in body composition between active and passive commuters to school. Fourteen studies observed that active commuters to school had a more favorable body composition and one study reported that ACS was associated with unfavourable body composition. ACS could be the steppingstone to improve PA promotion in adolescence but whether ACS is associated with improved body composition and prevention of obesity requires further research.

## 1. Introduction

The prevalence of overweight and obesity in youth aged 5 to 19 years has risen dramatically worldwide (from 4% in 1975 to over 18% in 2016) [1]. A recent study from Spain shows that the prevalence of childhood obesity has increased in the past two decades by 1.6% according to body mass index (BMI), and 8.3% according to abdominal obesity [2].

Many studies have evidenced the important role of physical activity (PA) to prevent and reduce the prevalence of obesity [3,4]. However, PA levels decline with age and adolescence is a critical period in predicting the development of obesity and medical morbidity during adulthood [5]. Therefore, it is important to promote PA during adolescence to ensure meeting PA guidelines over a long-term [5,6].

Recent studies have demonstrated active commuting to school (ACS) (i.e., travelling to/from school on foot, by bicycle or riding a scooter) is one of the means to increase PA levels in youth [7,8,9] and reduces the prevalence of obesity [10,11,12]. Many studies suggest that ACS has a positive impact on body composition parameters in obese and overweight children. These studies examined the association between ACS and weight status [10,11,13], BMI levels [13,14,15,16], fat mass (FM) values obtained by skinfolds measurements [13,14,16] and waist circumference [14,17]. However, other studies [18,19,20,21] and reviews [8,22,23,24,25,26] in children have shown inconsistent findings in association with ACS and obesity.

Faulkner et al. (2009) [25] in their review, reported that there was little evidence to suggest a relationship between ACS and healthier body weight/BMI among children. It was raised that maybe some aspects such as distance to school and the intensity engaged in those trips could be relevant in associative empirical studies examining the relationship between ACS and bodyweight/BMI. The same review outlined that there was a need to explore this topic in adolescents.

The walking school bus was very successful in Mexican adolescents to promote ACS and it has been shown to be a good strategy to decrease obesity [18]. However, cultural adaptation and pilot testing of these types of strategies should take place to understand their feasibility and acceptability in other countries.

Larouche et al. (2014) [26] also highlighted the need to evaluate the impact of existing programs that promote ACS (i.e., walking school buses, safe routes to school, and classroom based approaches) on PA levels and health related outcomes in adolescents.

Most of the previous research has been conducted in children [8,24,25], or youth (i.e., children and adolescents together) [22,23,26], but there is a paucity of research in adolescents only. No review articles or meta-analyses on the topic were found specifically in adolescents. Children and adolescents differ in obesity levels and their compliance with the recommendation of PA and eating habits [2]. PA levels decline as children move to adolescence and therefore adolescence is a critical period for interventions aiming to increase/maintain PA levels and reduce sedentary time in this age group. ACS provides adolescents with an opportunity to engage in regular PA which may help maintain and/or increase their PA and as a result assist with prevention of unhealthy weight gain during adolescence. To address this gap in knowledge, the aim of this review was to investigate the association between ACS and obesity parameters in adolescents.

## 2. Materials and Methods

The purpose of this review was assessing the breadth of scientific evidence linking ACS with overweight/obesity parameters in adolescents aged 11 to 19 years.

### 2.1. Information Sources and Search Strategies

For conducting a literature search we used PubMed, WOS and SPORTDiscus as electronic databases and applied search strategies using the following search key words: for the *population* (adolescent OR adolescence OR youth OR teen OR teenager OR children) AND *context* (active commuting OR active transport OR cycling OR walking) AND *outcome* (obesity OR overweight OR body composition OR fat mass OR body mass index) AND *school*.

The literature search was carried out from January 2021 to March 2022. From all articles resulting from the search, the abstracts were carefully read, and relevant studies were selected and reviewed in full. To determine a relevant study, at least two of the researchers (EMM, CR) had to confirm the validity of the theme of the article independently, in case of divergence, a third researcher (SA) was invited to decide whether to include or exclude the studies. Some authors (EMM, SA) participated in the analysis and interpretation of findings for each topic. A bias risk calculation was included in the process. All authors reviewed all paper (EMM, CR, SA, SM, AQ). Furthermore, an adaptation of the STROBE checklist was used [27] to evaluate the quality of the articles included in our study. STROBE checklist contains 15 items of analysis. The cut-off points established for quality classification were as follows: 0–5 points = low quality; 6–10 points = moderate quality; and 11–15 points = high quality. Studies that achieved high quality, according to the quality criteria, were included in the final analysis. These authors reviewed the whole process (EMM, CR, SA, AQ).

### 2.2. Study Eligibility Criteria

We only included: (a) studies written in English; (b) active commuting to school-based studies; (c) adolescents aged 11–19 years; (d) studies that included results on overweight/obesity measures; and (e) studies involving children and adolescents that performed analyzes stratified by age group. We did not include studies with no participants’ age specified or children with disabilities. Finally, we excluded from consideration, studies of other topics, duplicated studies, and protocols. All the steps of the process followed the recommendations of the PRISMA Flow Diagram [28], as presented in Figure 1.

### 2.3. Data Items

To classify commonality between articles, data were collected and organized by year of publication, author, study population and characteristics, study location, study design, method used to assess ACS and body composition and outcomes of measures.

## 3. Results

Figure 1 details all the steps of the process followed according to the recommendations of the PRISMA flowchart in the studies selection. We identified a total of twenty-two studies [29,30,31,32,33,34,35,36,37,38,39,40,41,42,43,44,45,46,47,48,49,50] published between 2003 and 2021 that presented data on the association between adolescents’ commute mode to school and overweight/obesity which were included in the systematic review from the original 513 articles identified by the literature search. The most common reason for excluding studies was that they did not meet the inclusion criteria, mainly by age.

Descriptive characteristics of these studies are presented in Table A1 (Appendix A). Seventeen studies were cross-sectional [30,31,32,33,34,36,37,39,40,41,43,44,45,46,47,48,50] and five had a longitudinal design [29,35,38,42,49]. According to the study population, the review comprised a total of 38,136 participants.

Studies included a large study population (345 to 7023 participants) and different geographical locations. Seven studies were from Europe [30,32,34,35,38,39,43], two from Africa [46,49], five from Asia [29,31,47,48,50], seven from the US [33,36,37,40,42,44,45] and one from Australia [41]. Significant relationships between BMI and ACS were found in all the US studies, and the same was found in Asia except for one study from the Philippines [29]. All studies from the US used the BMI *z*-score measure. In the European studies, three of them [30,34,35] found no relationship with obesity or overweight, while the other four [32,38,39,43] did. In Australia, no significant relationships between BMI and ACS were found. In Africa, one study [46] found significant associations, while another [49] did not. In addition, in South America, two [37,45] found associations and one [36] did not.

### Active Commuting and Obesity

From all studies included in this review, 15 articles (68.18%) [31,32,33,37,38,39,40,42,43,44,45,46,47,48,50] found a consistent association between ACS and body composition and seven studies (31.82%) [29,30,34,35,36,41,49] showed no differences in body composition between active or passive commuters to school. Fourteen studies [37,38,39,40,42,43,44,45,46,47,48,50] observed that ACS users had more favorable body composition than non ACS users and one study [33] reported that ACS was associated with higher average BMI.

To measure obesity levels, several variables have been used throughout the literature. Studies commonly used weight values (mainly direct measurements except one study which used self-reported measurements) to calculate body mass index (BMI). Other studies used skinfold measurements [32,36,40], bioimpedance analyses [32] and densitometry X-ray [49] to obtain FM values.

The strongest relationship between ACS and overweight/obesity status was found in two longitudinal studies [38,42] which used BMI to measure obesity levels. Drake, M. et al. (2012) showed that adolescents who walked/cycled to school more than 3.5 days per week were 33% less likely to be obese compared with adolescents who never walked/cycled to school. One of the weaknesses of that study was that they relied on height and weight self-report and proxy reporting by parents, which may have incorporated some possible bias; Bere, E. et al. (2011) showed that there were longitudinal associations between cycling to school and weight status. Furthermore, the overweight prevalence increased for those who stopped cycling to school compared to those ones who did not.

Three longitudinal studies [29,35,49] reported no significant relationships between ACS and prevalence of overweight/obesity status. Meron, D. et al. (2011) [41] reported that the short duration of walking trips (median 9 min) and possible slow pace undertook, may not have been sufficiently enough to impact upon weight status in adolescents. Therefore, it seems important to include a description of the type of route (i.e., duration, pace, distance) in the studies when analyzing the relationship between ACS and prevalence of obesity status. The three longitudinal studies [29,35,49] did not include such information.

It seems that the favorable relationship between ACS and overweight/obesity status depends on the frequency (i.e., number of times per week) of ACS, distance and time of the route to and from school [30,32,42,44].

In regards of the frequency, Drake et al. (2012) [42] reported that ACS frequency (measured by number of days per week) should be taken into consideration. In the US, attributable risk estimates suggested that obesity prevalence would decrease by 22.1% (95% CI: 0.1%, 43.3%) if all adolescents walked/biked to school at least 4 days per week [42].

In terms of the distance to and from school, short routes to/from school encourage children and adolescents to ACS and hence the improvement of their prevalence of obesity status [30,32] through a good sustainability of this behaviour. However, other authors outlined the need of a greater distance to be effective to reduce overweight/obesity levels [43]. Landsberg, B. et al. (2008) [32] reported that active commuting per se did not provide a sufficient amount of PA to affect adolescents’ FM or BMI, and distance to school should therefore be considered.

## 4. Discussion

The aim of this review focuses on the analysis of the relationship between ACS and overweight/obesity levels in adolescents only. After completing all the recommended procedures for systematic reviews (PRISMA flow-diagram and quality of articles checklist adapted from the STROBE instrument), 22 original articles were included in the analysis.

While 68.18% of the studies have shown that ACS was associated with more favorable body composition, the findings seem not to be consistent. Existing evidence [8,25,26] in a sample of children and adolescents together also reported these inconclusive results. These findings could be explained by several factors. One of them, showed that adolescents who reported cycling at least 1 h/week had additional health benefits, including lower BMI and waist circumference, however associations in levels of walking outcomes were inconsistent [51]. Therefore, the amount of PA seems to be very important in relation to have a positive effect in lowering obesity and overweight levels. On the other hand, Madsen et al. (2009) [33] and other recent study [52] corroborated that it is important to consider not only travel mode shift but also the obesogenic environment and unhealthy food/drinks purchases/consumption during adolescents’ school journeys, particularly in lower socio-economic areas, to prevent obesity.

On the other hand, other studies [53,54] have shown that for decreasing obesity levels in adolescents, ACS could only be part of a multicomponent approach, being as important as PA and diet. Therefore, it seems difficult to analyze the relationship between ACS and obesity levels alone. As we know, childhood obesity could be reduced by multi-component modifiable behavioral changes, including physical activity, but also sedentary behavior and dietary patterns [53]. Recently, the focus has shifted to following a more integrated approach in which it is recognized that 24-h movement behaviors are co-dependent. High PA, less sedentary behavior and sufficient sleep time is beneficial in maintaining a healthy body weight and reducing adiposity in children [53,54,55]. ACS could also be included as good strategy into the 24-h movement behaviors for either children or adolescents.

It has also been argued that the absence of significant differences in several studies could be due to the different body composition measures, and by not controlling the distance from home to school in the analyses. Therefore, in future research it is important to reference distance, frequency and duration of travel to/from school. In particular, Landsberg, B. et al. (2008) highlighted that ACS per se did not provide enough PA to affect adolescents’ FM or BMI and that distance to school should be considered for that purpose. For example, ACS was significantly inversely associated with overweight/obesity among adolescents who commuted beyond a one-half mile threshold [10]. Moreover, it is worth mentioning that distance of ACS is normally lower in children than adolescents [56]. Considering that adolescents can walk longer distances [41], it may be worth it to provide them with the education and strategies needed to use their ACS consciously as an opportunity to increase calorie expenditure and hence reduce their overweight/obesity status.

This review has strong aspects such a good quality report of the included studies. Moreover, this appears to be the first systematic review using the context of commuting to school that puts together, exclusively, adolescents (11–19 years) and body composition.

The present study has some limitations including a limited number of studies that examined ACS and body composition specifically in adolescents. Another limitation is the large heterogeneity in the measurement of body composition and ACS. Moreover, there is evidence that other factors such as social socioeconomic status and some lifestyle behaviors (sleep patterns, total physical activity, diet, chronotype and sedentary behaviors) [2,57,58,59] directly influence body composition and could bias the result of the studies.

## 5. Conclusions

The results obtained in this review suggest that evidence for the impact of ACS in promoting healthy body weights for adolescents is inconclusive. It seems that frequency of school journey trips, as number of days per week, and distance to/from school need to be included in future analyses. ACS could be a good strategy to increase PA levels in adolescence but, to improve body composition and prevent obesity we need further research. Moreover, cultural adaptation and pilot testing of some successful strategies should take place to understand their feasibility and acceptability in all countries. In addition, studies should consider the total amount of PA performed as well as the obesogenic environment and unhealthy food/drinks purchases/consumption during adolescents’ school journeys, particularly in lower socio-economic areas, to prevent obesity.

Moreover, our findings emphasize the need for more intervention studies in European adolescents, distinguishing by gender, age and population type (rural, urban and semi-urban areas) to increase the quality of evidence in the relation between ACS and obesity, with homogeneous measurements (BMI or fat mass). Self-reported measures of body composition should be avoided and, in addition to BMI, more anthropometric data such as abdominal circumference, percentage of fat mass and percentage of muscle mass are recommended.

## Figures and Tables

**Figure 1 ijerph-19-10852-f001:**
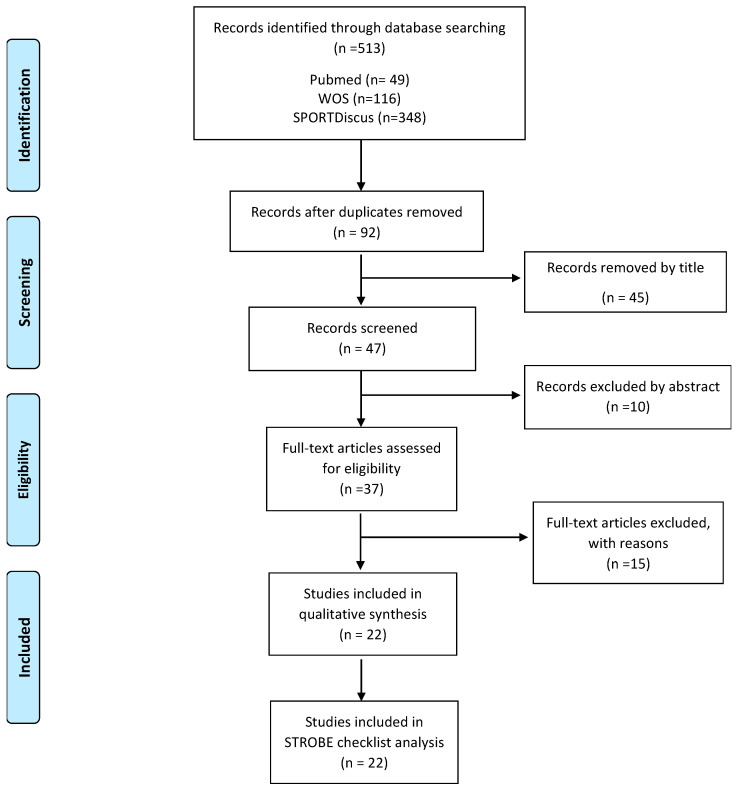
PRISMA flow-chart.

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
