# Peer review of "Associations among Active Commuting to School and Prevalence of Obesity in Adolescents: A Systematic Review"

_ijerph, 2022, doi:10.3390/ijerph191710852_

Round 1
Reviewer 1 Report
Concise discussion section and clear Appendix Table. Conclusions could be developed further with recommendations for future research (both design and measurement) and inclusion of age/gender/developmental maturation with additional factors that need study.
Author Response
We thank the reviewer for his/her comment, we feel the conclusions are clearer now. We have added more information in lines 228-244:
“The results obtained in this review suggest that evidence for the impact of ACS in promoting healthy body weights for adolescents is inconclusive. It seems that frequency of school journey trips, as number of days per week, and distance to/from school need to be included in future analyses. ACS could be a good strategy to increase PA levels in adolescence but, to improve body composition and prevent obesity we need further research. Moreover, cultural adaptation and pilot testing of some successful strategies should take place to understand their feasibility and acceptability in all countries. In addition, studies should consider the total amount of PA performed as well as the obesogenic environment and unhealthy food/drinks purchases/consumption during adolescents’ school journeys, particularly in lower socio-economic areas, to prevent obesity.
Moreover, our findings emphasize the need for more intervention studies in European adolescents, distinguishing by gender, age and population type (rural, urban and semi-urban areas) to increase the quality of evidence in the relation between ACS and obesity, with homogeneous measurements (BMI or fat mass). Self-reported measures of body composition should be avoided and, in addition to BMI, more anthropometric data such as abdominal circumference, percentage of fat mass and percentage of muscle mass is recommended.”
Reviewer 2 Report
There have been previous systematic reviews on the relationship between Active commuting to school (ACS) and obesity, but the authors' findings are interesting. I suggest authors should use PRISMA 2020 statement to evaluate the effects of ACS. Overall, this manuscript requires many improvements, Below are suggestions/indications with page and line indications:
Line 22 The "31.81%" should be 31.82%.
Line 73 The authors used STROBE checklist to select articles, but this work did not include in figure 1. PRISMA flow-chart.
Figure 1 The numbers in the flowchart are unclear, such as why did "SPORTDiscus (n=348)" disappear? Please check "RecorPubmed (n= 49)"
Line 221-228, Please check.
Line 236-361 The reference format should be carefully reviewed and corrected.
Author Response
We thank the reviewer for his/her time and effort. We answer to each of the comments individually below.
Reviewer: Line 22 The "31.81%" should be 31.82%.
Authors: we have corrected it. See line 22 and 137.
Reviewer: Line 73 The authors used STROBE checklist to select articles, but this work did not include in figure 1. PRISMA flow-chart.
Authors: We thank the reviewer to point this out. This information has now been included in Figure 1:
“Studies included in STROBE checklist analysis (n=22)”
Reviewer: Figure 1 The numbers in the flowchart are unclear, such as why did "SPORTDiscus (n=348)" disappear? Please check "RecorPubmed (n= 49)"
Authors: we thank the reviewer for this comment. We made a mistake, and it has now been corrected (see Figure 1)
Reviewer: Line 221-228 Please check.
Authors: We have included the contribution from authors in lines 245-251:
“Author Contributions: Conceptualization, S.A. and E.M.M.; methodology, E.M.M., S.A., S.M. and A.Q.; software, E.M.M.; validation, C.R-B., S.A. and A.Q.; formal analysis, E.M.M. and C.R-B .; investigation, S.A., S.M. and A.Q.; resources, E.M.M and S.A.; data curation, E.M.M., C.R-B. and S.A.; writing—original draft preparation, E.M.M., S.A.; writing—review and editing, E.M.M., C.R-B., S.A., S.M., A.Q. ; visualization, E.M.M., C.R-B., S.A., S.M., A.Q.; supervision, S.A., A.Q. and S.M.; project administration, S.A.; funding acquisition, S.A. All authors have read and agreed to the published version of the manuscript.”
Reviewer: Line 236-361 The reference format should be carefully reviewed and corrected.
Authors: we have corrected the reference format.
Reviewer 3 Report
The background description of the article is not clear enough, the necessity and scientificity are insufficient, and it lacks the necessary content of many literature review articles, such as forest plots, sensitivity analysis, publication bias and other data and chart descriptions. Therefore, it is believed that the article failed to meet the publication requirements, and it is recommended that the overall work be revised before submitting to other journals.
Author Response
We thank the reviewer for his/her time and effort. Several paragraphs have been included to clarify the rationale for the study in the background section of the manuscript. We thank the reviewer for his/her comments, we feel we did not clarified well enough the fact that in the adolescent population , there is a need to research in this topic and, to have a review of the evidence, will help to design future intervention studies. Please see lines 49-61:
“Faulkner et al.(2009)[25] in their review, reported that there was little evidence to suggest a relationship between ACS and healthier body weight/BMI among children. It was raised that maybe some aspects such as distance to school and the intensity engaged in those trips to school could be relevant in associative empirical studies examining the relationship between ACS and bodyweight/BMI. The same review, outlined that there was a need to explore this topic in adolescents.
The walking school bus, was very successful in Mexican adolescents to promote ACS and it has been shown to be a good strategy to decrease obesity [18]. However, cultural adaptation and pilot testing of these types of strategies should take place to understand their feasibility and acceptability in other countries.
Larouche et al. (2014)[26] also highlighted the need to evaluate the impact of existing programs that promote ACS (ie, walking school buses, Safe Routes to School, and classroom based approaches) on PA levels and health related outcomes in adolescents.”
See lines 64-65:
“No review articles or meta-analyses on the topic were found specifically in adolescents.”
See lines 66-71:
“PA levels decline as children move to adolescence and therefore adolescence is a critical period for interventions aiming to increase/maintain PA levels and reduce sedentary time in this age group. ACS provides adolescents with an opportunity to engage in regular PA which may help maintain and/or increase their PA and as a result assist with prevention of unhealthy weight gain during adolescence.”
Reviewer 4 Report
The reviewed manuscript is well written. The editor can consider the manuscript for this journal. The selective comments towards the authors are:
[1] The manuscript needs one-two to illustrate figures. It can be the relation between the body weight and ACS or different heights and ACS.
[2] Is there any variation in geographical location and ACS? For example, Asia and Europe or Africa are different than the US.
Author Response
We thank the reviewer for his/her time and effort.
Reviewer: The manuscript needs one-two to illustrate figures. It can be the relation between the body weight and ACS or different heights and ACS.
Authors: We thank the reviewer for his/her input. If we understand the reviewer’s comment correctly, the reviewer seems to be requesting figures that illustrate the relationship between the body weight and ACS or height and ACS. Given that at least some of the weight status classifications for adolescents are determined taking into account age and gender as well as body mass index (which includes both body weight and height measurements), we believe that it would not be appropriate to provide the requested figures even if body weight and height data were available from the published articles.
Reviewer: Is there any variation in geographical location and ACS? For example, Asia and Europe or Africa are different than the US.
Authors: we thank the reviewer for this comment, we have found this analysis very useful. We have added more information in lines 128-134:
“Significant relationships between BMI and ACS were found in all the US studies, and the same was found in Asia except for one study from the Philippines[29]. All studies from the US used the BMI z-score measure. In the European studies, three of them [30, 34, 35] found no relationship with obesity or overweight, while the other four [32, 38, 39, 43] did. In Australia, no significant relationships between BMI and ACS were found. In Africa, one study [46] found significant associations, while another [49] did not. And in South America, two[37, 45] found associations and one [36] did not.”
Round 2
Reviewer 2 Report
Dear authors
Thank you for considering my suggestions and incorporating them into the manuscript.
Congratulations for the research and keep up the good work.
Reviewer 3 Report
This article has been supplemented as requested.